# Learning Styles Determine Different Immigrant Students’ Results in Testing Settings: Relationship Between Nationality of Children and the Stimuli of Tasks

**DOI:** 10.3390/bs9120150

**Published:** 2019-12-10

**Authors:** Sandra Figueiredo, Tânia Brandão, Odete Nunes

**Affiliations:** 1Department of Psychology and Sociology, Universidade Autónoma de Lisboa Luís de Camões (UAL), Rua Santa Marta, Palácio Dos Condes Do Redondo 56, 1169-023 Lisbon, Portugal; tbrandao@autonoma.pt; 2Department of Psychology and Sociology and I&D CIP―Psychology Research Centre, Universidade Autónoma de Lisboa Luís de Camões (UAL), Rua Santa Marta, Palácio Dos Condes Do Redondo 56, 1169-023 Lisbon, Portugal; onunes@autonoma.pt

**Keywords:** auditory input, visual stimuli, ethnic groups, second language, school learning, learning styles, personality differences

## Abstract

Background: Literature presents little examination on the learning styles and sensorial preferences of immigrants during decoding of different tasks in testing contexts. Methods: In this cross-sectional study, non-native children (between 2nd and 12th grade) were divided into six groups determined by country of origin and examined on different stimuli, visual and auditory, associated with four tasks that measure cognitive and linguistic specific abilities. Results: The multivariate analysis confirmed that the children’s nationality significantly explained achievement variability regarding picture recognition and auditory discrimination. *η*^2^ values indicated that there were moderate to larger effects for the nationality as a factor that explains the variance of performance. Conclusions: Results indicate that tasks’ stimuli can effectively assess and differentiate specific young minority groups in order to understand their actual level of preparation and their needs for further learning. The listening input, on the one hand, should be established as the main differentiator for all groups at the time of school entry, but, on the other hand, it should be avoided in Asian groups and Eastern European students during the first stages of second language (L2) learning in European contexts with romance languages as the target learning.

## 1. Introduction

The verbal stimuli as presented in second language (L2) tests could influence the results of those groups and explain their maladaptive behaviors concerning academic failure. The stimuli and typology of tasks administered to different minorities should be seen as cues to understand their academic and cognitive behaviors (cognitive mapping and coding). Visual stimuli (either by means of pictures or orthographic cues) and auditory stimuli are differently encoded with regard to the students’ age of acquisition (of L2) [1]. L2 auditory discrimination is perceived as critical in the first years of acquisition of a second language, irrespective of other variables such as the proficiency already obtained and the socioeconomic background (SES) [1]. Even considering that balanced or unbalanced bilinguals have a high inhibitory control system for auditory input during listening comprehension [2], there is no consistent knowledge to understand if all L2 learners (unbalanced bilinguals) develop the same control systems with regard to their specific group of origin. This lack of knowledge indicates the need to have guidelines to evaluate the minorities at schools. According to the “Guidelines for research in ethnic minority communities” of the American Psychological Association (APA) [3], the ethnic groups, focusing on immigrants in the US, must be examined and studied taking into account their language of origin, their culture and group meaning (the case of the islanders), their migration experience, and individual acculturation preferences. These preferences are related to the tests’ administration with previous cross-cultural verification [3] and ethnic group combination (Chinese students are different from the Asian islanders). On the other hand, these guidelines for psychological tests’ administration and research might be adapted for the academic and cognitive evaluation involving ethnic groups. Additionally, the research results should be analyzed concerning the groups’ culture of origin to avoid misconduct in the research and evaluation of ethnic populations. It should also be pointed out that the most significant studies conducted about immigrant specific minorities focus on American contexts and there are misunderstandings of minorities acculturation regarding their host country as a main variable [3].

The standardization of measures regarding academic performance and cognitive ability became a practice that underestimates the skills of the minorities. The common variables in research studies with immigrant populations and in European contexts [4], such as socioeconomic status, should be examined with covariates [5]. Other factors might emerge to explain the cultures’ reactions to certain stimuli in the tests’ approach. The results of some groups depend on their linguistic and cultural background. They require cognitive strategies in a distinct manner and the individuals’ text mapping is determinant for the resolution of sound or visual coding [6,7,8]. Graham [9] concluded that listening comprehension tasks for second language learners are especially difficult because L2 learners self-evaluate their skills as very basic in the listening area. The comprehension of information coded in sound files can be a major challenge, and also a test cue to differentiate minorities and to enhance different cognitive and linguistic skills. On the other hand, the listening decoding in unfamiliar language is improved when the visual cues (such as pictures) are presented alongside the auditory input and thus assist the discrimination task [10,11,12]. The research has focused on the study of mental processes (how they report strategies used in real time during listening tasks) involved in listening decoding [13]. However, there are no studies that examine that difficulty among minority groups. Graham [9] maintained that speed, decoding of blocks or individual words, and new vocabulary are the main obstacles during listening comprehension, but also that they are the most common steps used by decoders. The text-based strategy is characterized as a bottom-up process, different from the top-down process, which is context-based. The bottom-up method for decoding in L2 was determined to be the cognitive strategy most used by learners and to be related to their ability and exposure to listening tasks. 

About the discrepancy observed in performance both in listening and in image (pictures) decoding tests, as well as regarding the conflict between top-down and bottom-up strategies [14,15], the learning style should also be seen as an important variable [16,17,18]. There are three main sensorial preferences: visual, auditory, and kinaesthetic. Early [19,20] and more recent [21,22] studies have highlighted that different cultures and nationalities, along other variables like age, determine the learning style in the foreign language. Likewise, visual and auditory inputs are preferences observed in good readers against poor readers’ sensory preferences for language learning. However, these results are adjusted to the native populations. Non-native individuals attain their optimal learning if the teaching community understands the minority’s differences concerning learning styles [23]. In another line of studies, learning styles are examined as a possible match to tasks and strategies for effective academic learning [24]. This research seeks to understand how the learning style affects visual and auditory stimuli in evaluation tests administrated to immigrant young students. Asians, (only assessed in the US considering their sensory preferences during L2 learning) as being less participative in the classroom, predetermined by their cultural values and educational habits in their prior schools, in their countries of origin, it is likely that the same minority prefers visual than auditory stimuli. If Asians do not participate in classroom activities, as compared with their non-native peers, their phonological skills and phonetics are too low to achieve high scores in listening tasks. 

Visual inputs, through pictures or orthographic information, are expected to be suitable for faster learning. As said previously, specific minorities may involve other cultural groups considering their countries of origin, such as the case of Asians. Chinese L2 learners behave differently from Iranian learners regarding their learning styles [25]. However, age and gender affect learning styles [21,26,27]. Nationality, age, and gender should be considered when calculating the correlation between learning styles and second language learning. Visual learning styles are more prominent for language learning, for instance, with regard to vocabulary retention [28] On the contrary, the auditory preference for language learning is constrained when the individual has limited vocabulary in the target language. The teaching styles (methodologies and strategies administered at classroom) affect also the learning process of different immigrant learners [29] and this matter should be explored in further studies to understand correlation between teaching and learning (involving teachers at regular classroom and the immigrant young students).

The main question of this study is established in this correlation: how are the visual (picture and orthographic cues) and auditory inputs in tests related to the decoding skills of learners of Portuguese as L2 when taking the immigrant school groups as the main moderator? This research intends to examine whether there are differences between ethnic and immigrant populations when performing specific tasks with pictures, written words, and auditory exercises as stimuli for language and verbal reasoning assessment in Portuguese as a second language. Taking into account the reaction to each stimulus, we may obtain the necessary information of immigrants’ skills in second language as well as their cognitive preferences toward tests. We hypothesized that immigrant students, according to their cultural background and prior school experience, react differently to tests based on visual or auditory stimuli. Previous studies indicate that Asians and Latinos tend to experience more academic failure and language difficulty in the L2 domain, and that Asian immigrants and Hispanics performed less in auditory comprehension against visual input decoding. Given the few studies conducted in European contexts, we hypothesized that European immigrants (from other countries in Europe) in Portugal would have higher rates of decoding both for visual and auditory input. Vocabulary proficiency (both for L2 and for L1) is assumed to influence success in auditory discrimination tasks, along with the cultural influence of students when responding to pictures or sounds. These two factors are expected to significantly diverge and interact to explain the performance of specific and diverse immigrant groups. 

## 2. Materials and Methods

### 2.1. Participants

Upon arrival, the young immigrants are assessed using a proficiency test with no norm reference (schools have autonomy to use proficiency tests that need to be based on the European Common Framework) [5], but they are not taught according to differentiated learning methods (considering native and non-native students merged at classroom) and they are not identified based on different evaluation methods. After the diagnostic proficiency test, immigrants and refugees are enrolled in regular classes with or without parallel support programs to help their inclusion in school. Those programs focus mainly on language learning and evaluation in specific academic disciplines when children are deemed to be underperforming. In Portugal, the education system requires 12 years of compulsory education. All students involved in this study attended 11 Portuguese schools in the district of Lisbon, from year 3 until year 12. Out of the 108 immigrant students, 81 did the recall test (two levels: recall of full text and words, in writing); 98 did the picture-based phonemic task; 93 children did the picture naming task; 89 did the dichotic hearing test. Children below A1 proficiency level did not carry out the tasks because they were not eligible for these specific tasks. In the core sample, 46 (43%) were male and 59 (55%) female, with a mean age = 13 years old (SD = 2.7). Through a Chi-Square analysis, the distribution of the age groups was not significant (*p* > 0.05) according to the country of origin. Twenty-five students came from China, 6 from Latin America, 31 from Eastern Europe (mainly Russia and Ukraine), 19 from Portuguese-speaking African countries, 12 from Western Europe and 14 from the Indian subcontinent. The mean length of residence (LOR) or age of acquisition (AOA) was 3–4 years. The groups originating in China, other Asian countries and Africa were the last to arrive in Portugal. The sample included students whose proficiency in Portuguese was up to B2 level, the maximum allowed to conduct the study. The main home languages were: Mandarin, Russian, Ukrainian, African creoles (Portuguese lexical basis), Indo-Aryan Languages, Spanish, and French. In most cases, the children had had prior schooling experience in their countries. The families had the same several languages’ background as their children. As for socio-economic status, participants were evaluated taking into account the educational background and the professional situation of their families (employed or not employed in Portugal; profession with or without specialization: positions that require or not having an academic degree). Only the Chinese group of students received home language instruction (Mandarin) alongside the Portuguese school instruction. The L1 instruction took place on Saturdays during the school year. 

### 2.2. Materials

The subjects completed the four tasks during the full test battery implementation that took place in the classroom, in an environment suited to the research sessions, over two years. The four tasks comprised two different stimuli and two different support mechanisms: pictures on paper and auditory files in the computer. All tasks were intercalated by visual and auditory tests according to the order they were administered. The first pair of tasks consisted of a story recall test (1) (computer) and a picture naming test (2) (paper). The second pair of tasks had a picture phonemic recognition task (3) (paper) and a dichotic listening test (4) (computer). The tasks involving listening input were intended to measure the auditory level of discrimination correlated to the proficiency evaluation focusing on words and full text. The students were asked to write words and text as listed (1 and 4) and as observed in pictures (2 and 3). The tasks were as follows:

(1) The story recall test was developed to examine the attention and memory of L2 learners after listening to a short text. This test was adapted from the story recall tasks of Woodcock-Muñoz language survey-revised [10]. Portuguese text was inserted and the answer was required in writing, different from the original Woodcock’s task. Children were asked to recall as many texts as they could, respecting the order of events in each text. For word recalling there was no need to write in order of appearance. The students were instructed to do the following: to follow a processing order in the decoding task: to listen to the texts; to recall the texts (1) and words (2) without second access to the audio source; then write the recalled text considering the order of information recalled. The full story recalled had a score of 2 points. As for only words, 1 point was attributed for each correct word recalled. 

(2) The picture naming task was administered according to the original from the *Diagnostic Test for Portuguese as a Non-Native Language* [30], with 36 images presented sequentially over five pages, in paper format. Participants answered in writing only with nouns for all the images listed. The picture naming task evaluated the children’s ability to use useful and descriptive vocabulary. The images were arranged in a low level vocabulary frequency [31] and referred to general objects in the family, school, and social domains. Fully correct answers to this task had a total score of 12 points. 

(3) The picture phonemic recognition task was originally developed for a previous research project [8] and included three levels of phonemic evaluation: the syllable, the onset, and rime. Pictures were used only for the decoding task and answers on phonemes’ units. The syllable level of this task is only reported here because it resulted in significant scores that distinguish the ethnic groups of the participants. The main goal of this tasks item is to assess the phonological awareness at intrasyllabic level by using images and not orthographic cues. The pictures evoke familiar referents (“rat”, “cherry”, “duck”, “shoe”, belt”) and answers were required in writing. Fully correct answers were awarded a total of 3 points.

(4) The dichotic hearing test had 2 items presented in a binaural mode. Information was sent to each ear. The individuals used the computer and headphones provided for the test. During the hearing exercise they should pay attention to words in Portuguese and nonwords. The participants were not informed of the nonwords presence in the audio file. This task examines listening ability as well as vocabulary, memory, and attention. It was hypothesized that individuals may frequently encode the nonwords in Portuguese words similar in orthographic form. The transfer events and the conversion of nonwords into words retrieved in their L1 were expected to occur. After the listening, participants were instructed to write down the answers. Each correct answer (words or nonwords correctly spelled) scored 1 point, up to a maximum of 8 points (4 words per ear, of which 2 were nonwords). 

### 2.3. Procedure

In all four tasks, the children were introduced to the tasks in a double manner: computer and paper simultaneously. The full battery of tests took more than 60 minutes on average. According to Woodcock [10], more than seven tasks for second language assessment should be longer than 45 min. For the specific block of the four tasks, individuals took approximately 20 min. They were informed of the tasks’ procedure and carried them out during their classes. The schools were also informed about it during the authorizations process. The children’s families were also informed about the evaluation goals and the involvement of their children in the tests. Participants were controlled for nationality and home language. The tests were administrated according to spaced intervals. Other tasks, mostly written, were also administrated along with this study’s four tasks.

All subjects gave their informed consent for inclusion before they participated in the study. The study was conducted in accordance with the Declaration of Helsinki and the protocol was approved by the Ethics Committee of FCT—Fundação para a Ciência e a Tecnologia under Grant no. SFRH/BPD/86618/2012.

### 2.4. Data Analysis

The participants’ answers were coded according to the score assigned for each task. Incorrect or incomplete answers were not considered. The performance of the groups’ nationality was examined using multivariate analysis of variance. Partial *η*^2^ is reported as a measure of effect size to determine the significant effects. Partial eta squared was preferable to present the real results reported and not claimed data (eta squared) [32]. All statistical tests were determined at *p* < 0.05 level using SPSS, version 24 (IBM SPSS Inc., Chicago, IL, USA). For the sample’s description, frequency statistic tests were carried out and to identify relationships between age, gender, nationality, and length of residence (date of arrival to Portugal), chi-square tests were performed. The tasks were not compared between themselves as to their different scores (example: naming tasks with 36 points if fully correctly answered and part of the recall test depends on the number of correct spelt words). Levene’s test was performed to evaluate the normality criteria of the sample. Tasks with statistical significant values (*p* < 0.05) concerning the sample distribution were tested with Kruskal-Wallis independent test for samples. 

## 3. Results

In the first part of the statistical analysis, we carried out multivariate analyses of variance to assess how significantly the nationality groups differ in the stimuli along the four tasks. As a first procedure, the results were run with a verification test for assumptions to analyze variance and covariance. Homogeneity of variance was tested (Levene’s test) to inform about the use of parametric tests. Considering the non-homogeneity for variance of sample (*p* < 0.05) observed only in the picture naming task, the Kruskal-Wallis test for independent samples was used. The univariate analyses of variance and the Kruskal-Wallis test showed that the children’s performance in the four tasks showed significant differences in the six groups of immigrants. The statistical differences were significant in all children’s groups in the auditory discrimination and visual decoding tasks: (*p* > 0.05) auditory text recall (a) (F(5,80) = 5.414, *p* = 0.03, partial η2 = 0.285) and (b) (F(5,80) = 4.129, *p* = 0.02, partial η2 = 0.179), picture naming (F(5,93) = 2.319, *p* = 0.04, partial η2 = 0.117), picture phonemic recognition (F(5,98) = 25.24, *p* = 0.3, partial η2 = 0.219) and dichotic hearing (words transfer) (F(5,83) = 2.730, *p* = 0.025, partial η2 = 0.108). The Manova allowed to compare means and standard deviation error of performance in all tasks. η2 values indicated that there were moderate to larger effects for the nationality as a factor that explains the variance of performance along the four tasks by comparing the six groups of students. The nationality moderating effect was more evident in the performance for text recall than in the results for words’ recall (the same story recall task).

The post-hoc results, through *Tukey* test adjustment, showed specific significant contrasts between groups. The results indicated that Chinese L2 learners have more problems decoding in the tasks with auditory input-recall and dichotic hearing, than in the visual input tasks. In these, the Portuguese Asian and the Eastern European children had the worst scores. For the recall task (a) the Chinese group performed worse compared to the other groups but only significantly different (*p* < 0.05) from the Europeans (from Eastern countries) and from the other Asian group (Indian subcontinent). These three groups had the lowest means despite the fact that Chinese learners were the worst performers. For the recall task (b) Chinese L2 learners differed significantly from the Latinos and the Western Europeans. These two last groups had the highest scores in this task. Still concerning the auditory discrimination, the dichotic hearing showed significant differences (*p* < 0.05) in the groups, mainly regarding the conversion of words to other similar lexicon (both in Portuguese or L1). On the other hand, there were two specific nonwords that participants decoded, differing significantly among nationalities, as nonwords. Both Asian groups had the lowest means against the higher scores of the Europeans. There were specific words with statistical significant decoding during the dichotic task among the different groups: words “bola” (‘ball’) and “jaula” (‘cage’) and decoding of nonwords “risga” and “leta”. However, the different words were distinctly perceived by Chinese and Latinos in a different way. The Chinese had the lowest means of correct answers. We observed that words and nonwords with more open vowels generate higher significance during decoding in all groups of learners.

Regarding the visual discrimination tasks-naming and phonemic recognition-with pictures as the main input for decoding in Portuguese as L2, the results showed a different scenario than that observed in the auditory listening tasks. The Indian subcontinent (the other Asian group) performed worse when compared mainly to the Eastern Europeans with the highest means (Table 1). The naming task was examined using the Kruskal-Wallis non-parametric test and the null hypothesis (groups did not differ in performance for naming task) was rejected.

In the same multivariate analysis of variance, through covariate analysis, the interaction of Nationality × Condition Age did not produce a significant result for the tasks’ performance. On the other hand, the parents’ nationality was introduced as a new interaction to the comparison of means analysis of variance and, even when no significant alterations were observed, the results remained significant and high concerning the type of stimuli in the tasks, mainly high for the auditory stimuli (partial *eta squared* increased: 0.319). In general, the European children (Westerners) showed higher means compared to the Asian children. We concluded that performance was more influenced by nonverbal prior knowledge (strategies acquired during L1 development and also strategies taught at their prior school in their respective countries) than by language proficiency in Portuguese. Definitely, performance was not influenced by prior language storage since the four tasks focused on bottom-up strategies and were aimed at learners with limited proficiency. 

## 4. Discussion

This study intends to further the comprehension about the measurement tasks for immigrant children as students at risk of experiencing academic failure in the host schools. Four tasks to evaluate language levels’ awareness (lexicon and phonemic) and associated verbal reasoning (recall of prominent episodes and their order as well as the main keywords, lexicon transfer, dichotic decoding through words and nonwords, syllable counting, and word-image match) were administrated to second language learners, immigrants, in Portuguese schools. Each pair of tasks had different stimuli for language decoding: visual (pictures) and listening (using words and nonwords) inputs. The multivariate analysis results indicated that the Asian children, both Chinese and from the Indian subcontinent, had the lowest scores compared to Western European children. However, these differences and low performance levels were not equally distributed among the children’s minorities. The task inputs (stimuli) appeared to influence performance. Chinese children performed the worst when the tasks demanded listening comprehension while they did not show statistically significant differences and did not have the lowest means in the visual input tasks. Concerning visual decoding, the other groups contrasted: Asian (not Chinese) had the lowest means and the Eastern European had the highest performance. However, the Eastern European children also had more difficulty in the auditory recall task. Regarding the dichotic input specific auditory task, both Asian groups had the lowest scores, as opposed to the European one (Westerners). Part of the results support the data of previous studies referring to immigrant young students in the US, which identify the Asian group as the one most at risk of academic and psychological distress [33,34,35]. The word recognition tasks showed results that prove the instability that non-Western students (immigrants) are experiencing when they are coding and decoding in very different systems (concerning the writing and the noun-verb paradigm in Portuguese). Host schools are not preparing the language skills and verbal reasoning in a well manner for the different minorities if the teachers and scholars have little knowledge about the home languages and cognitive mapping of their immigrant students.

Referring to the cognitive load, the listening input processing involves bottom-up more than top-down strategies, which had a facilitating effect on novice L2 learners because it processes the instant auditory stimuli rapidly [36,37]. Also, bottom-up strategies, less text-based, and low proficient strategies are more related to the cognitive and neural mapping commonly used in complex word and orthographic cues processing [38]. In the first years of L2 learning, interacting with the adaptation psychological process, the auditory stimuli could affect the L2 learning [39]. The socioeconomic status and general background of these immigrants can block auditory discrimination in L2 [1]. Portuguese immigrant children when arriving at school have different levels of preparation for the current syllabus and the current referenced diagnostic proficiency tests. These tests are probably not educationally adjusted to the different minorities and ethnic groups. For the auditory and visual inputs, memory and attention are involved differently to successfully complete these tasks. Non-native learners fail more significantly in the listening tasks due to attention deficits [39] and to the high difficulty of tasks, such as dichotic decoding [40]. We conclude that auditory discrimination requires more attention skills than picture discrimination when not using Stroop tests. However, this cognitive requirement is more difficult for specific groups of non-native individuals. Their abilities, as healthy individuals, such as attention and memory, are not compromised, but they are failing due to specific input decoding in specific target languages such as the Portuguese.

According to this specific empirical study, the Asian, mainly the Chinese, struggle the most with tasks that involve auditory information other than visual. Age is not responsible for the groups’ difference in those tasks, nor is the age of acquisition, as normally observed [41,42,43]. These results suggest that the initial diagnostic tests of the skills of immigrant students when joining the host school should differ in their input. Auditory tasks should be used in children from the European, American and African continents, and visual tasks should be used in children from Asian countries, mainly from China. In addition, the tasks in the educational programs for second language and academic general skills learning should take these differences into account. Visual and auditory discrimination should be differently included for minorities, enhancing their solving problems in the areas they find more difficult: auditory discrimination in L2 for Chinese and visual (picture and orthographic) decoding for Asian from Indian countries. Further studies should also consider the dual task (picture and listening stimuli simultaneously) in the evaluation [13,44] of differences among the minorities, regarding the younger ages, to understand how to mitigate the auditory discrimination difficulty, especially in specific groups of immigrants. As for listening comprehension and concerning context-based strategies, there are several abilities to be developed in immigrant school population, such as auditory discrimination, working memory [13], audience awareness [45], and prior L1 knowledge (in terms of conceptual meaning, not only linguistic). Regarding the visual comprehension involved in decoding images and sequence of images, there are other abilities involving visual memory, prior cultural background, and prior schooling experience. The early stages of L2 learning are critical in terms of tasks and their stimuli [44]. The main question is to determine how different the tasks need to be for distinct cultural backgrounds in the same classroom.

## 5. Conclusions

Further studies should also examine immigrant students considering their learning styles as evaluated at their country and school of origin, mostly studies of immigrated minorities that reveal higher levels of difficulty in academic and social adaptation: different Asian groups and the Hispanics. This study stresses that auditory discrimination tasks, more than visual discrimination tasks, with words and nonwords as stimuli, could improve the differential diagnostic test for the specific minorities in European schools. Auditory tasks are indicated for children from the European, American and African continents and visual tasks are more suitable for children from Asian countries, mainly from China. These are contributions for a set of guidelines to be developed for a new adjusted model more in accordance with the diversity of immigrant school population. Those guidelines may replicate the guidelines for evaluation of minorities and ethnic groups established by the APA [3]. According to the APA guidelines and to other studies [32] on ethnic minorities, the Hispanic population is frequently at risk in terms of educational needs and development. However, in this study the Hispanic population did not have low scores and thus, it is not part of an educational risk group in Portugal. The current generation of immigrants and refugees in contexts other than the American require new reference frameworks that go beyond language assessments.

## Figures and Tables

**Table 1 behavsci-09-00150-t001:** Second language learners’ performance in different tasks determined by auditory and visual stimuli.

Tasks	Nationality		Parents’ Nationality	
	*F*	^1^ partial *η*^2^	M	%
Auditory stimuli (recall test–full text)	5414	0.285	1649	0.119
Auditory stimuli (recall test–words)	4129	0.179	1503	0.119
Visual stimuli (picture naming)	2319	0.117	1937	0.130
Visual stimuli (picture phonemic recognition)	2524	0.219	1312	0.088
Auditory stimuli (dichotic listening)	2730	0.108	1662	0.126

^1^ Eta value (*η*^2^) ranging from 0.01 to 0.06 refers to smaller effect, between 0.06 and 0.14 refers to moderate effect, >0.14 refers to larger effect.

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
