# Peer review of "Learning Styles Determine Different Immigrant Students’ Results in Testing Settings: Relationship Between Nationality of Children and the Stimuli of Tasks"

_behavsci, 2019, doi:10.3390/bs9120150_

Round 1
Reviewer 1 Report
I would like to suggest to widen and deepen the introduction of the article and to better contextualize the setting and the research question.
Very good presentation of participants and materials. I find the description of the data analysis a little too concise. Interesting discussion and conclusion.
Author Response
Firstly, many thanks for the refereeing process, for the very positive evaluation and for the suggestions addressed (minor revisions) in order to improve our paper. Concerning all the corrections and suggestions, changes were made (lines and length were added in the text, properly). All the commentaries were considered and they are all answered and clearly indicated (identifying line and pages when necessary) below. The all paper was checked considering grammar and language by the international office affairs of our University.
Attending the two points to be considered by the Reviewer 1, here the confirmation of the corrections performed:
point 1 - The introduction of the article was developed in more deep analysis and with better contextualization. At first, we did not developed this Introduction attending the length, but now please see text and authors added from line 67 until line 117 (new authors were indicated for contextualization and deepen arguments and settings). As regarding the suggestion advanced in this Report, also the Research Question was developed accordingly: lines 118 - 135.
point 2 -The data analysis section was improved from line 238 until 240 (the scores were explained for the tasks as well examples).
Reviewer 2 Report
This is an important study that addresses increasingly prevalent, though comparatively less well-examined, questions of immigrant and migrant children learning experiences and opportunities and related questions of education in multicultural and post-colonial contexts. This research is clearly crucial for developing culturally relevant and appropriate methods and materials.
No English, writing, technical and stylistic issues.
Introduction and literature review indicate the need for such a study although the literature review can be improved. In particular, the citation style makes it difficult to evaluate the relevance and rigor of the sources used to bolster foundational assumptions.
Some attention to history would go a long way to making the work more rigorous. For example, it could briefly address, through secondary sources, the legacy of cognitive mapping and its complicity, intended or otherwise, in stigmatizing and marginalizing socio-cultural and ethnic difference. This problem has occurred and has been studied in other places with high degree of immigration coupled with colonial legacies, so some mention of these factors and history in Portugal, even if only to dismiss the question, seems necessary.
A related potential shortcoming, alluded to in places, is that if and when culturally relevant teaching is shown improve learning, how it might be integrated into curriculum and teaching and learning on day to day basis. This could be at least noted as for future work.
Given the multi-cultural make-up of the subjects, the method prioritizes noun recognition and somewhat overlooks non-Western cultural norms. Better attention to differences between noun and verb based oriented languages, or the role of oral based and text based languages, as well as the related questions of the metaphorical use of language in general that consider relational thinking, would go a long way toward giving making the study more convincing from the standpoint of socio-cultural and historical studies of these often overlooked issues in schooling and education.
Similarly, a somewhat more thorough explanation of the auditory and visual perception differences can be helpful, given this being a key aspect of the findings.
Author Response
Firstly, many thanks for the refereeing process, for the positive evaluation from the two Reviewers and for the suggestions (and important corrections) addressed in order to improve our paper. Concerning all the corrections and suggestions, changes were made. All the commentaries were very well considered and they are all answered and clearly indicated (identifying line and pages) below.
Point 1 - Introduction and literature review were effectively improved (as mentioned for the Report 1) as well the citation style was clarified (all the format for the references in text is corrected now; several were added and the length of the paper was changed). Please see the changes/new text properly due in lines 118-135 and 238-240.
Point 2 - Regarding the cognitive mapping and the ethnic minorities, authors were added in order to increase and assure the foundation of our arguments in the paper: mapping and cognitive strategies are different indeed considering ethnicities and linguistic minorities. Portuguese immigrants are example of that reality but very few studies focused the diferences of cognitive strategies and mapping across minorities in Portugal. Considering another populations we placed new info in lines 67-117 and new authors for example in line 73.
Point 3 - The mention to the teaching and teachers' Styles and the cultural diferences to be correlated with teaching/learning process (indeed very important despite of not being our focus in this specific paper) was inserted as referred for future work (lines 114-117). In fact we have a published study focusing this matter: teaching styles and their correlation with different ethnic sensorial preferences fotr learning.
Point 4 - The noun recognition was outlined as a limitation and also a focus of this study in lines approx. around 268/269, in the Results section. The word recognition was a problem for the immigrants with Indo-Aryan Languages (and indian subcontinent) which proves that there are linguistic and ethnic groups (due to their first language, prior school knowledge and home writing system) which face difficulties with coding/decoding in Western context (considering the language and the school system). All these was outlined in the Results. Also this explanation and observation were included and clarified in lines 315-320. More references also added.
Point 5 - Yes, all clarified and more developed text in Introduction for the auditory and visual perception diferences (for example: 116-135=.
Many thanks again for the efforts and for the improvement granted within this study!
Sandra Figueiredo
(corresponding author).